# Safety and Effectiveness of Shoulder Arthroplasties in Spain: A Systematic Review

**DOI:** 10.3390/jcm8122063

**Published:** 2019-11-23

**Authors:** Jorge Arias-de la Torre, Xavier Garcia, Kayla Smith, Arantxa Romero-Tamarit, Elisa Puigdomenech, Laura Muñoz-Ortiz, Jonathan P. Evans, Vicente Martín, Antonio J. Molina, Carles Torrens, Miquel Pons-Cabrafiga, Francesc Pallisó, Jose María Valderas, Mireia Espallargues

**Affiliations:** 1Agency for Health Quality and Assessment of Catalonia (AQuAS), 08005 Barcelona, Spain; xavier.garciacusco@gencat.cat (X.G.); ksmith@gencat.cat (K.S.); arantxa.romero@gencat.cat (A.R.-T.); epuigdomenech@gencat.cat (E.P.); lmunyoz@gencat.cat (L.M.-O.); mespallargues@gencat.cat (M.E.); 2CIBER Epidemiology and Public Health (CIBERESP), 28029 Madrid, Spain; vicente.martin@unileon.es; 3Institute of Biomedicine (IBIOMED), University of Leon, 24071 León, Spain; ajmolt@unileon.es; 4King’s College London, Institute of Psychiatry, Psychology and Neuroscience (IoPPN), London SE5 8AB, UK; 5Health Services Research on Chronic Patients Network (REDISSEC), 28029 Madrid, Spain; 6Health Services and Policy Research Group, University of Exeter Medical School, Exeter EX1 2LU, UK; J.Evans3@exeter.ac.uk (J.P.E.); J.M.Valderas@exeter.ac.uk (J.M.V.); 7Royal Devon and Exeter NHS Foundation Trust, Exeter EX2 5DW, UK; 8Hospital del Mar, Department of Orthopaedic Surgery and Trauma, 08003 Barcelona, Spain; ctorrens@parcdesalutmar.cat; 9Department of Orthopaedic Surgery and Trauma, Sant Rafael University Hospital, 08035 Barcelona, Spain; 23655mpc@gmail.com; 10Department of Orthopaedic Surgery and Trauma, Santa María University Hospital, 25198 Lleida, Spain; fpalliso@gss.scs.es

**Keywords:** shoulder, arthroplasty, safety, effectiveness, systematic review

## Abstract

The effectiveness and safety of shoulder arthroplasties in the general context of a Spanish patient population remains unclear. The aim of this study was to ascertain both the effectiveness and safety of primary shoulder arthroplasties and the prosthesis types used in Spain. A systematic review of all the available literature evaluating the effectiveness and safety of primary shoulder arthroplasties in Spain was performed. A narrative synthesis was performed, and evidence tables were created in four dimensions: study design, arthroplasty characteristics, safety, and effectiveness. Orthopaedic Data Evaluation Panel (ODEP) scores were used to evaluate prosthesis types. Twenty-one studies were selected that included a total of 1293 arthroplasties. The most common indication was fractures, while the prosthesis most frequently used was the Delta Xtend (ODEP 10A). The most common complication was scapular notching. Prosthesis revision rate was approximately 6% for follow-ups between 12 and 79 months. In addition, significant improvements were observed in the Constant–Murley test score after the intervention. Currently in Spain, shoulder arthroplasty can be considered a safe and effective procedure with functional recovery and pain reduction for eligible patients with humeral fracture, rotator cuff arthropathy, fracture sequelae and malunion of the proximal humerus, and degenerative disease. Future longitudinal research and population-based studies could serve to confirm these results and identify points of improvement.

## 1. Introduction

Shoulder arthroplasty is currently considered to be an established therapeutic option and an effective and efficient procedure to improve physical function, pain and quality of life in patients [1,2,3,4,5]. As a result of continual technological progress and emerging indications for shoulder arthroplasty, including proximal humeral fractures, osteoarthritis and massive rotator cuff tears [6], the utilization of this procedure has increased throughout the world and in some countries it has tripled in the last decade [7,8,9].

The use of shoulder arthroplasty has increased significantly in Spain in recent years [10,11], but remains a lower-volume procedure compared to knee and hip replacement. Given the increasing indication for these types of procedures and their potential to improve the health of patients, it is vitally important for both patients and clinicians that the complications associated with these interventions are well understood. Previous research has proposed that the three most common complications are instability, periprosthetic fracture and infection [12,13,14]. Understanding this information within the context of the effectiveness of different types of prostheses and models, and in certain population groups, is likely to be highly relevant [1,7,14]. However, as far as we know, no studies have evaluated the effectiveness and safety of these procedures in the general context of a Spanish population. Furthermore, there is no arthroplasty register that can be directly assessed. Both analysing the results of these procedures by population and establishing a registry could be useful in evaluating the results of shoulder arthroplasties more precisely in a specific healthcare context such as Spain, facilitating a comparison to other international contexts.

Regarding the safety and effectiveness of primary shoulder arthroplasties, evidence suggests that some of the most frequent complications associated with this procedure are scapular notching, dislocation, periprosthetic fracture, and infection [14,15,16]. Furthermore, the revision rate of shoulder arthroplasties is estimated to be approximately 5% and 10% at 5 and 10 years respectively, and may be lower in reverse arthroplasties compared to hemiarthroplasty and total shoulder arthroplasty [15,16,17]. However, in the Spanish context, to date no population-level studies have been conducted to evaluate these outcomes, or aimed to quantify the safety and effectiveness of primary shoulder arthroplasties. Furthermore, the evidence found in specific studies that have already been carried out is divergent, which may be due to a focus on specific models or types of prostheses, or certain pathologies and clinical populations [18,19,20].

In this context, the objective of this study was to describe the scientific evidence available on the effectiveness and safety of primary shoulder arthroplasties in Spain and the types of prosthesis used in this population.

## 2. Materials and Methods

A systematic review was conducted on the results of shoulder arthroplasties performed in public hospitals in Spain, and the results are reported according to the Preferred Reporting Items for Systematic Reviews and Meta-Analyses (PRISMA) criteria [21]. The review has been registered in PROSPERO (CRD42019115342). The following databases were used as sources of information: EMBASE, PubMed, Scielo, Cochrane Reviews and Center for Reviews and Dissemination. The search range was restricted from January 2003 to December 2018. Given the continuous improvement in shoulder arthroplasty results and the advancement in surgical techniques, the lower limit was set at 2003. This limit was fixed in order to maximize the study period and overcome any potential limitations related to underestimating the current results after pooling long-term retrospective data.

A search filter was developed specifically designed for PubMed/Medline to achieve the objectives of this study (Appendix A), and was adapted to other databases. The search strategy was based on previous studies in an attempt to maximize the number of documents identified [22,23,24]. Keywords for procedure as well as anatomical and territorial location were used. In addition, the references found in systematic reviews and meta-analyses were used to identify primary studies, a grey literature search was conducted, and key authors were contacted.

### 2.1. Inclusion and Exclusion Criteria and the Revision Process

The PICO criteria (population, intervention, comparison, outcome) were used to identify studies. We included documents in English and Spanish that focused on the evaluation of effectiveness and safety in primary shoulder arthroplasties performed in public hospitals in Spain. Due to limitations related to the robustness of the data and possible biases when making inferences in the population, only studies with a sample size of 20 or more primary interventions were included. Documents that included patients under 18 years of age, studies that evaluated revision implants or those indicated for tumours or congenital diseases, and studies aimed at evaluating surgical techniques were excluded. Additionally, studies evaluating complications, adverse effects and/or effectiveness based on certain patient characteristics were excluded due to the difficulty in generalizing and comparing their results.

A screening of the title, summary and full text was carried out independently by two expert reviewers (JAT and XGC), while possible discrepancies were resolved by a third reviewer (KS). After study selection, a narrative synthesis of the evidence obtained was carried out. Given the variability in study characteristics and outcome variable presentation, a meta-analysis was deemed unfeasible. Therefore, the information was extracted in various tables of evidence with four dimensions: study design, arthroplasty characteristics, safety and effectiveness of primary shoulder arthroplasties. To evaluate the prosthesis models, the Orthopaedic Data Evaluation Panel (ODEP) scores were used [25]. ODEP is a panel of independent experts that publishes reference indexes to assess the effectiveness of different models of anatomical and reverse prostheses. The criteria used were based on implant survival, follow-up time and the size of the cohort analysed. Based on these criteria, implants were assigned to categories in terms of the evidence supporting their use.

### 2.2. Quality of Studies Included

The quality of the studies and their design were considered according to the Scottish Intercollegiate Guidelines Network (SIGN) levels of evidence hierarchy [26]. Furthermore, the Risk of Bias tool (RoB 2.0) was used for randomized trials to assess the quality of evidence [27], while for non-randomized trials, the Risk of Bias in Non-Randomized Studies of Interventions tool (ROBINS-I) was used [28]. Finally, for single-cohort study designs, the scale of evidence assessment for case series studies from the Institute of Health Economics was used [29].

SIGN levels were assigned according to the quality of the study evidence based on their design from the 1 ++ level of higher evidence reserved for high-quality meta-analyses, systematic reviews of randomized clinical trials, and randomized clinical trials with very low risk of bias, to level 4, which includes expert opinion. To compare studies, the risk of bias for each reference was calculated using the RoB 2.0 and ROBINS-I tools [27,28], assigning 4 points to “critical” risk assessments; 3 points to “high” risk; 2 points to “moderate” risk, “some considerations” or cases assessed with “insufficient information”; and 1 point for “low” risk assessment. Next, the percentage of points obtained over the total points possible for all categories was calculated. For single-cohort studies, the percentage of positive responses was obtained from the assessment scale.

## 3. Results

A total of 360 references were identified (259 from EMBASE, 84 from PubMed and 17 from SCIELO) (Figure 1). After eliminating duplicates, 323 were screened by title and abstract. After screening, 277 documents were excluded, thus including 46 for full-text review. Of these 46 references, 25 were excluded and 21 studies were included in the final evidence tables [18,19,20,30,31,32,33,34,35,36,37,38,39,40,41,42,43,44,45,46,47].

Table 1 shows the level of evidence of the selected studies. Two randomized clinical trials had a SIGN 1+ level and two non-randomized clinical trials a SIGN 2 ++. The remaining studies were observational, 11 of which were retrospective. In terms of risk of bias assessment, little variability was observed in the risk attributed to the different studies. The greatest risk of bias came from subjectivity in measuring outcome variables, followed by the possible impact of uncontrolled confounding variables. The average range for patient follow-up was between 12 and 79 months, and included the results of 1293 arthroplasties. The number of cases per study ranged from 21 to 163.

Table 2 shows that the average age of patients included was over 70 years, approximately 70% of whom were women. In addition, two studies documented low comorbidity in their patients (Charlson Comorbidity Index less than 2) [18,19]. There were four main indications for undertaking primary arthroplasty: acute fractures and fracture-dislocations of the proximal humerus, rotator cuff arthropathy, fracture sequelae and malunion of the proximal humerus, and degenerative diseases. About half of the interventions were performed using a deltopectoral approach, while the rest used a superolateral or anterosuperior approach. In terms of implant characteristics and fixation, 18 studies included reverse prostheses, 8 of which used a cemented fixation and 6 of which were non-cemented. The most frequently used prosthesis models included in these studies were the Delta Xtend, with an ODEP assessment of 10A, and the Delta III, which is not evaluated by ODEP. The third most frequent was the Lima SMR, with an ODEP rating of 10A.

Table 3 shows that the most frequent complication in reverse shoulder arthroplasties was scapular notching, reported in 14 studies [33,48]. One study found a higher presence in older patients [46]. In terms of complications associated with tuberosities, malunion was documented in six studies, with the maximum rate being 33%. Additionally, resorption of the tuberosities was reported in five studies. Prosthesis infection was documented in seven of the included studies, with one study citing up to 8% [34]. Intraoperative fractures were also documented in seven studies, with the highest values being 7%. Similarly, periprosthetic fracture in five documents and ossification in one were also seen. There was a 1% dislocations rate, and less than 1% were complications relating to fixation, positioning, and movement of the prosthesis. The main neurological vascular and lymphatic system complications were paralysis in three studies and hematoma in three others. In terms of prosthesis survival, 67% of the studies selected cited approximately 6% revision rate between 12 and 78 months. A significant difference was found in the revision rate in two studies comparing partial arthroplasties (hemiarthroplasty) to reverse arthroplasties [32,34]. The revision rate for hemiarthroplasties was far higher, with a difference greater than 15% in both studies.

In terms of the effectiveness of shoulder arthroplasties, Table 4 shows that in the studies evaluating patients before the intervention, the Constant–Murley test score was approximately 30%. After the intervention, the average score was approximately 65%, with significant improvement reported in four studies [20,30,34,41]. Seven of the eight studies included metrics before and after reported improvement in external rotation, with three studies being statistically significant. In one study, better results in terms of hemiarthroplasties were observed in reverse prostheses [32]. Five of the seven studies that dealt with internal rotation and analysed metrics before and after reported improvements in movement. One of these studies had a significant difference. Better results were observed in patients that had fractures with tuberosity union compared to a group of patients with tuberosity malunion [35]. Sixteen of the 21 studies considered other results, highlighting their frequency on the UCLA Shoulder Rating Scale (UCLA) in two studies, Disabilities of the Arm, Shoulder and Hand (DASH) or QuickDASH in six studies and on Visual Analog Scale (VAS) in four studies.

## 4. Discussion

The results of the review show that shoulder arthroplasty in Spain can currently be considered an effective and safe procedure, with functional recovery and pain reduction in patients operated on for humeral fracture and rotator cuff arthropathy, fracture sequelae and malunion of the proximal humerus, and degenerative diseases. These results are similar to those found in other countries, including Norway, Germany, the Netherlands and the USA [9,15,49,50,51], with better results observed with reverse-type arthroplasties than hemiarthroplasties.

Regarding the safety of shoulder arthroplasty, the overall complication rate in this procedure appears to be centred around 15%, with the most commonly observed complications being instability, periprosthetic fracture or infection [13,14]. The results obtained from this study show that the most frequent complications in Spain were of the same profile as other countries, with similar rates also reported [9,15,52,53,54]. In addition, increased safety has been observed in recent years worldwide as reported by the Nordic or the Kaiser Permanente registers [15,55]. These improved results could be for various reasons, with progress in prosthesis design being particularly relevant. The presence of reverse prostheses should be noted as they have become one of the treatments of choice for pathologies like proximal humeral fractures or rotator cuff arthropathy [56,57,58,59]. The results obtained when assessing reverse prostheses suggest that, while their rate of complications might be slightly higher than those observed in other contexts [9,15,52,53,54], their results could be better when compared to other types of prostheses [32,33,36,48]. This facilitates the hypothesis that reduced incidence of the aforementioned complications could be largely due to an increased use of these types of implants.

Currently, the evidence for implant survival in Spain at the population level is limited. However, the results of the reviewed studies are similar to data from international registries, which estimate an implant revision rate of approximately 90%–95% at 5 and 10 years [9,15,49,50,52]. In addition, most of the prosthesis models identified in this review are commonly used internationally, and the ODEP assessment of most of those included in the selected studies was acceptable [9,25]. Similar to our results, the 10-year cumulative revision rate after primary reverse shoulder arthroplasty in the Nordic countries was between 90% to 95% and the model most frequently used was Delta Xtend [55]. Moreover, the results from other European shoulder arthroplasty registers, including the National Joint Registry (NJR) in the United Kingdom and the Dutch arthroplasty register (LROI) show that the results in terms of survival rates might be similar across European countries and that the prosthesis models most frequently used are usually the same [51,60]. However, considering the results for shoulder arthroplasty effectiveness, in terms of functionality, pain and impact on the patient’s life, there is some difference in calculating scores, which hinders synthesis [61,62]. Regardless of the calculation differences, especially for the Constant–Murley test, improved scores after the intervention were observed in the results of the studies reviewed. The results show that an improvement in pain could be relevant, even at short-term follow-ups after arthroplasty, which is contrary to results suggested by a previous study proposing that improvement in the short-term may not be as evident as in longer-term follow-ups [63].

The authors accept that a significant limitation of the present study is the search strategy. Given its focus on the Spanish population, extrapolating and generalizing its results to other populations is challenging. However, we believe that delimiting the safety and effectiveness of shoulder arthroplasties in a specific healthcare context may be useful in encouraging and improving results at all levels: surgical, management, and patient. In addition, the results shown can be useful and relevant at the international level when making comparisons and establishing common standards of reference. It is also important to mention the limitation related to the inclusion criteria, given that patients under 18 years old and patients with tumours were excluded. These criteria restrict the capacity to extrapolate the results obtained to the whole population. Despite this limitation, the results found could be applicable to most of patients eligible for a shoulder arthroplasty, and thus it is reasonable to assume that they could be at least approaching the true results of these procedures in Spain. Another limitation is related to the heterogeneity of the studies in terms of their design and presentation of results, which makes a meta-analysis impossible. However, the evidence presented was synthesized as much as possible to be able to serve both as a reference in assessing the safety and effectiveness of shoulder arthroplasty, and as a starting point for new studies on the subject. Lastly, these studies were considered without stratifying by patient diagnosis or the severity of their symptoms. As such, it is possible that some of the results described correspond to selected samples of patients with certain diagnoses. Despite this, most of the studies focus on the two main reasons for intervention in shoulder arthroplasty: fracture of the humerus and rotator cuff arthropathy, which is why we believe the results shown can be widely generalized to the population susceptible to receiving a shoulder arthroplasty and not only to those with one of the two diagnoses. However, further epidemiological research stratified by these indications both in Spain and in other countries could be valuable to obtain a more precise representation of the safety and effectiveness of shoulder arthroplasties at the population level.

## 5. Conclusions

In Spain, primary shoulder arthroplasties, for those who are able to receive them, are an effective and safe procedure that allow functional recovery and pain reduction in patients with humeral fracture, rotator cuff arthropathy, fracture sequelae and malunion of the proximal humerus, and degenerative diseases. The prosthesis type with the best survival is the reverse prosthesis. Future longitudinal population-based studies, particularly randomized controlled trials, as well as the establishment of a shoulder arthroplasty registry could confirm these results and identify areas of improvement, including the recommendation of specific types of prostheses or models with preferable results.

## Figures and Tables

**Figure 1 jcm-08-02063-f001:**
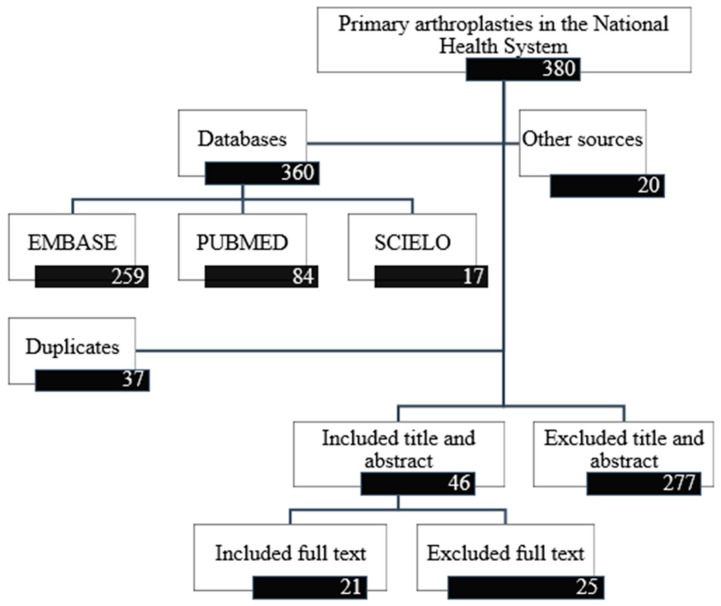
Study selection flow diagram.

**Table 1 jcm-08-02063-t001:** Primary shoulder arthroplasties. Evidence, risk of bias, and study design.

Author, Year	SIGN Level of Evidence	Risk of Bias (%)	Study Type (Time Period)	Average Patient Follow-Up in Months	Total Number of Cases
Torrens et al., 2016	1+	56	Randomized controlled trial (2010–2012)	24	81
Sebastià Forcada et al., 2014	1+	56	Randomized controlled trial (2009–2011)	30 reverse and 28 partial	61
Boyer et al., 2017	2++	39	Nonrandomized trial (2009–2011)	15 reverse and 25 partial	134
Alentorn-Geli et al., 2014	2++	50	Nonrandomized trial (2005–2012).	40 reverse and 72 partial	32
Jorge-Mora et al., 2018	2+	43	Retrospective observational (2012–2017)	26	114
Sebastià Forcada et al. 2017	2+	54	Retrospective observational (2009–2011)	40 plate failure and 37 acute fracture	60
Lopiz et al., 2016	2+	54	Retrospective observational (2009–2012)	33	42
Bonilla et al., 2012	2-	54	Retrospective observational (2003–2011)	25 Delta and 9 Comprehensive	43
Izquierdo-Fernández et al., 2017	2-	46	Prospective observational (2012)	48	29
Cáceres-Sánchez et al., 2015	3	50	Retrospective observational (2004–2012)	36	52
Martinez et al., 2012	3	43	Prospective observational (2003–2007)	48	44
Zafra et al., 2014	3	43	Prospective observational (2003–2011)	51	35
García-Fernandez et al., 2015	3	39	Retrospective observational (2003–2014)	79	163
Alcobía-Díaz et al., 2017	3	43	Retrospective observational (2009–2011)	53	116
Torrens et al., 2016	3	43	Prospective observational (NS)	12	60
Torrens et al., 2017	3	36	Prospective observational (NS)	24	58
Torrens et al., 2018	3	36	Retrospective observational (2010–2012)	29	41
Delgado-Rodríguez et al., 2013	3	54	Retrospective observational (2006–2010)	17	40
Hernández-Elena et al., 2015	3	43	Retrospective observational (2009–2013)	18	37
Villodre-Jiménez et al., 2016	3	43	Prospective observational (2008–2014)	34	30
Andrés-Cano et al., 2014	3	43	Retrospective observational (2009–2010)	21	21

SIGN: Scottish Intercollegiate Guidelines Network; NS: Not specified.

**Table 2 jcm-08-02063-t002:** Characteristics of shoulder arthroplasties included in the selected studies.

Author, Year	Diagnosis	Patient Characteristics	Characteristics of the Intervention	Arthroplasty and Fixation	ODEP Model Rating
Torrens et al., 2016	Rotator cuff arthropathy (82% G38 vs. 63% G42), Proximal humeral fracture (18% G38 vs. 26% G42) and fracture sequelae (13% G38 vs. 11% G42).	Average age, 75 years G38 (88% women) vs. 76 G42 (84% women).	Glenoid component fixed without retroversion, metaglene, flush. Deltopectoral approach for sequelae fractures and anterior superior in acute fractures, and pretension of the rotator cuff. Movement starting at 24 h, Sling: 3 weeks.	Reverse.	Delta Xtend - 10A; glenoid component size (38 vs. 42 mm).
Sebastià Forcada et al., 2014	Proximal humeral fracture in three (16% reverse vs. 13% partial) and four fragmented (84% reverse vs. 87% partial). Rotator cuff arthropathy: 55% reverse vs. 63% partial.	Average age: 75 years, reverse (87% women, 61% lesion in dominant arm) vs. 73 years, partial (83% women).	Deltopectoral approach. Sling: 3 weeks. Average time between fracture and surgery: 5 days. Rehabilitation: beginning of active and passive movement at 2 weeks, active with resistance at 6 weeks.	Reverse vs. partial (non-cemented).	Lima SMR - 10A.
Boyer et al., 2017	Three- and four-part proximal humeral fractures.	Average age: 78 years, reverse vs. 68 years partial.	Deltopectoral (88%) or superolateral (12%) approach. Average time between fracture and surgery: 7 days.	Reverse vs. partial. Non-cemented (using two screws).	-
Alentorn-Geli et al., 2014	Fracture sequelae of the proximal humerus.	Average age: 79 years, reverse (80% women) vs. 83 years, partial (33% women).	Deltopectoral (84%) or superolateral (16%) approach. Tuberosity osteotomy type IV. Subscapularis repair in partial arthroplasties and in the reverse deltopectoral approach.	Reverse vs. partial.	Reverse - Delta Xtend - 10A vs. Partial: Global Advantage.
Jorge-Mora et al., 2018	Proximal humeral fracture.	Average age: 78 years, cemented (100% women) vs. 76 years, non-cemented (91% women)/76 years union (95% women) vs. 78 years non-union (95% women).	Approach: deltopectoral cemented (92%) vs. non-cemented (94%)/deltopectoral union (92%) vs. nonunion (95%), the rest superolateral. Average time between fracture and surgery: 9 days cemented/8 days union vs. 8 days nonunion.	Total reverse: cemented vs. non-cemented.	Arrow shoulder fracture and anatomic shoulder reconstruction vs. Humelock II.
Sebastià Forcada et al., 2017	Complex fracture sequela due to fixation failure of proximal humeral plate vs. proximal humeral fracture.	Average age: 73 years, sequela (63% women, 43% dominant arm) vs. 75 years, fracture (63% women).	Deltopectoral approach. Average of 2.311 days between fracture and intervention in the group with sequelae. Sling: 3 weeks. Rehabilitation: started at 3 weeks, lasted 4 weeks.	Total reverse, non-cemented.	Lima SMR - 10A.
Lopiz et al., 2016	Humerus fracture: three (12% vs. 19%) and four (42% vs. 48%) fragmented and dislocated fractures (46% vs. 37.5%).	Over 80 years old vs. under 80 years old. 80% women. Dominant shoulder inured (62%).	Deltopectoral approach. Average of 6 days between fracture and intervention. Rehabilitation: passive movement from 24 h post intervention to 2 weeks, exercises for 3–4 weeks.	Reverse cemented.	Delta Xtend - 10A.
Bonilla et al., 2012	Rotator cuff arthropathy or osteoarthritis secondary to rotator cuff tear.	Average age: 76 years, Delta vs. (87.5% women, 81% right arm) vs. 72 years Comprehensive (92% women)	Approach: Delta Xtend transdeltoid vs. Comprehensive deltopectoral.	Reverse.	Delta Xtend - 10A vs. Comprehensive Reverse Shoulder System - 5A.
Izquierdo-Fernández et al., 2017	Rotator cuff arthropathy, fractures, or dislocations.	Average age: 78 years (80% women). Body mass index ≤35 vs. >35.	-	Reverse.	Delta Xtend - 10A.
Cáceres-Sánchez et al., 2015	Proximal humeral fracture (10%), prosthetic revision (12%), fracture sequelae (19%), rotator cuff tear (60%).	Average age: 70 years (84% women).	Deltopectoral approach. Sling: 3 weeks. Rehabilitation: passive movements between the 1st and 3rd weeks, active movements between the 2nd and 4th weeks and muscle enhancement between the 3rd and 12th weeks.	Reverse.	Delta Xtend - 10A (58%) and Aequalis Reversed - 5A (42%).
Martinez et al., 2012	Fracture sequelae of the proximal humerus.	Average age: 77 years (60% women).	Deltopectoral approach: Average time between fracture and surgery: 365 days. Rehabilitation: starting at 3 weeks.	Reverse cemented (45%) or non-cemented (55%).	Lima SMR - 10A.
Zafra et al., 2014	Error in the treatment of proximal humeral fractures in two (40%), three (26%), and four parts (34%).	Average age: 69 years. Fracture in dominant arm (86%).	Deltopectoral approach.	Reverse cemented.	Delta III.
García-Fernandez et al., 2015	Rotator cuff injury (30%), rotator cuff arthropathy (44%), proximal humeral fractures (26%).	Average age: 76 years (87.5% women) for the sample that included 40 additional revision arthroplasties.	Approach: deltopectoral (fractures) or superolateral (rotator cuff tears and arthropathies)	Reverse (cemented and non-cemented).	Delta III (9%), Delta Xtend (43%) - 10A, Lima SMR (29%).
Alcobía-Díaz et al., 2017	Rotator cuff arthropathy.	Average age: 81 years (88% women) Charlson Comorbidity Index: 1.7 = low comorbidity.	Superolateral approach (76%) or deltopectoral (24%). Passive rehabilitation during hospital stay, exercises for 6 weeks.	Total.	-
Torrens et al., 2016	Rotator cuff arthropathy.	Average age: 74.5 years (92% women).		Reverse.	Delta Xtend - 10A.
Torrens et al., 2017	Reverse (71%): 43% rotator cuff arthropathy, 28% proximal humeral fractures. Total: 22% primary osteoarthritis. Partial: 7% proximal humeral fractures.	Average age: fractures, 74 years; osteoarthritis, 78 years; arthropathy, 74 years (88% women).	Reverse: anterosuperior approach. Anatomic: deltopectoral approach. Average time between fracture and surgery: 11 days.	Total anatomic (23%), partial (7%) and reverse (70%).	Reverse: Delta Xtend - 7A, Total: Global AP - 5A, Partial: Global Unite.
Torrens et al., 2018	Proximal humeral fracture in three (17%) or four parts (83%).	Average age: 78 years (87.5% women). Average BMI: 28.	Anterosuperior approach. Average time between fracture and surgery: 12 days.	Reverse cemented.	Delta Xtend - 10A.
Delgado-Rodríguez et al., 2013	Proximal humeral fracture.	Average age: 76 years (87.5% women).	Rehabilitation: average start at 4 weeks. Average number of sessions = 39.	Partial.	
Hernández-Elena et al., 2015	Proximal humeral fracture with risk of osteonecrosis of the humeral head. Type of fracture: four fragments (54%), three fragments (30%), fracture-dislocation (16%).	Average age: 77 years (97% women).	Deltopectoral approach. Rehabilitation: starting at 2 weeks with passive movement, resistance exercises starting at 6 weeks.	Reverse.	Aequalis^®^ Reversed II - 5A.
Villodre-Jiménez et al., 2016	Humerus fracture in three parts (27%) and four parts (73%). With non-reconstructible fractures, risk of avascular necrosis, severe osteoporosis and previous rotator cuff injuries.	Average age: 75 years (87% women).	Deltopectoral approach. Sling: 3 weeks. Rehabilitation: passive movement starting at 3 weeks, exercises starting at 6 weeks.	Reverse cemented.	Lima SMR - 10A.
Andrés-Cano et al., 2014	Proximal humeral fracture in three (10%) or four parts (57%), fracture-dislocation (33%).	Average age: 72 years (90% women). Charlson Comorbidity: 0 or 1.	Deltopectoral approach. Average time between fracture and surgery: 17 days. Sling: 5 weeks. Rehabilitation starting at 4 weeks.	Partial non-cemented.	Epoca Shoulder Arthroplasty System.

ODEP: Orthopaedic Data Evaluation Panel; G38: 38 mm glenosphere; G42: 42 mm glenosphere; %: percentage; BMI: body mass index; -: no information available.

**Table 3 jcm-08-02063-t003:** Reliability of primary shoulder arthroplasties.

Author, Year	Scapular Notching	Infection	Fractures, Tears and Ossifications	Complications in Tuberosities	Fixation, Dislocation or Stiffness Complications	Neurological, Vascular and/or Lymphatic Complications	Reinterventions and Revisions
Torrens et al., 2016	Scapular neck notching (with intention to treat) 46% G38 vs. 30% G42.	0% G38 vs. 3% G42	-	-	Dislocation: 3% G38 vs. 2% G42.	-	Revision: 3% G38 vs. 2% G42.
Sebastià Forcada et al., 2014	Scapular neck notching: 3% Reverse vs. 0% partial.	3% reverse vs. 3% partial.	Intraoperative fracture: 0% reverse vs. 3% partial. Ossification: 16% reverse vs. 20% partial	Malunion: 19% reverse vs. 13% partial (*p* = 0.4). Resorption: 16% reverse vs. 30% partial (*p* = 0.4).	Rigidity: 0% reverse vs. 3% partial. Migration: 0% reverse vs. 20% partial. Radiolucency: 13% reverse vs. 10% partial.	Hematoma: 0% reverse vs. 4% partial.	Revision before the 40th month: 3% reverse vs. 20% partial.
Boyer et al., 2017	Notching: Reverse (8%) vs. partial (0%).	-	Periprosthetic fracture: 3% reverse vs. 1% partial. Cuff tear: 0% reverse vs. 5% partial.	Lysis 6% reverse vs. 2% partial.	Poor fixation: 7% reverse vs. 2% partial.	Phlebitis: 3% reverse vs. 0% partial. Paralysis: 1% reverse vs. 1% partial. Lymphedema: 0% reverse vs. 2% partial.	Revision: 7% reverse vs. 2% partial.
Alentorn-Geli et al., 2014	Glenoid erosion: 0% reverse vs. 8% partial.	0% reverse vs. 8% partial.	-	-	-	-	Revision: 0% reverse vs. 25% partial.
Jorge-Mora et al., 2018	-	Early infection: 0% cemented vs. 3% non-cemented.	Periprosthetic fracture: 0% cemented vs. 3% non-cemented.	Poor reinforcement: 33%, 54% cemented vs. 76% non-cemented (*p* = 0.07).	Dislocation: 0% cemented vs. 3% non-cemented.	Paralysis: 0% cemented vs. 3% non-cemented.	Revision: 0% cemented vs. 9% non-cemented.
Sebastià Forcada et al., 2017	0%.	-	Acromion fracture: 3% sequela vs. 0% acute. Intraoperative fracture: 0% sequelae vs. 3% acute.	-	Dislocation: 7% sequela vs. 0% acute. Loosening 3% sequela vs. 0% acute. Radiolucency: 7% sequela vs. 0% fracture.	-	Revision: 13% sequela vs. 0% fracture.
Lopiz et al., 2016	Scapular notching: 14%.		Periprosthetic fracture: 2%.	Malunion: 19%. Resorption: 5%.	Dislocation: 2%. Radiolucency: 0%.	Hematoma: 4%.	Revision: 2%.
Bonilla et al., 2012	Scapular notching: 31% Delta vs. 9%. Comprehensive.	6% Delta vs. 0% Comprehensive	-	-	Migration: 3% Delta vs. 0% Comprehensive.	-	Revision: 9% Delta vs. 0% Comprehensive.
Izquierdo-Fernández et al., 2017	Scapular notching: 47% BMI < 35 vs. 50% BMI > 35.	-	-	-	Radiolucency: 57% vs. 37.5% (*p* = 0.3).	-	-
Cáceres-Sánchez et al., 2015	Scapular notching: 17%.	4%.	Acromion fracture: 2%. Intraoperative fracture: 2%.	-	Radiolucency: 2% Instability: 4% Loosening: 4%.	-	Reintervention: 9%.
Martinez et al., 2012	Glenoid notching: 41%. Scapular notching: 41%.	-	-	Resorption: 7%.	Radiolucency: 11%. Dislocation: 14%. Loosening: 2%.	Paralysis 1%.	Revision: 11%.
Zafra et al., 2014	60%.	-	Intraoperative fracture: 5%. Periprosthetic fracture: 2%.	-	Radiolucency: 65% humeral, 31.5% glenoid.	-	Other complications (20%).
García-Fernandez et al., 2015	-	-	Periprosthetic humerus fracture: intraoperative with non-cemented Lima SMR (1%), postoperative with Lima SMR (1%).	-	-	-	-
Alcobía-Díaz et al., 2017	-	-	-	-	-	-	10% analgesic treatment 6 weeks after the intervention.
Torrens et al., 2016	-	-	-	-	-	-	-
Torrens et al., 2017	-	0%.	-	-	-	-	-
Torrens et al., 2018	Scapular notching: 15%.	-	Presence of osteophyte: 12%.	Malunion: 32%. Resorption: 10%. Poorer union, depending on comorbidity.	-	Paraesthesia: 15%.	Revision: 2%.
Delgado-Rodríguez et al., 2013	-	-	-	-	-	-	-
Hernández-Elena et al., 2015	Scapular notching (29%). Relationship between notching and age.	-	Intraoperative fracture: 2%.	-	-	Hematoma: 5%. Neuropraxia: 3%.	-
Villodre-Jiménez et al., 2016	Scapular notching (46%).	-	Intraoperative fracture: 7%.	Malunion: 33%.	-	-	Other complications: 46%.
Andrés-Cano et al., 2014	-	-	-	Malunion: 5%. Resorption: 24%.	Radiolucency: 5%.	-	-

G38: 38 mm glenosphere; G42: 42 mm glenosphere; BMI: Body Mass Index; *p*: *p*-value; -: no information available.

**Table 4 jcm-08-02063-t004:** Effectiveness of primary shoulder arthroplasties.

Author, Year	Constant–Murley Score	Joint Assessment, by Constant–Murley Score	Other Results
Torrens et al., 2016	Global score: 29 before—57 after G38 vs. 26 before—55 after G42. Pain: 5 before—11 after G38 vs. 5 before—11 after G42. Daily activities: 8 before—14 after G38 vs. 7 before—14 after G42.	Flexion = 4 before vs. 7 after G38 vs. 3 before—7 after G42. Abduction = 4 before vs. 6 after G38 vs. 3 before—6 after G42. External rotation = 2 before—5 after G38 vs. 2 before—4 after G42 (among groups after, *p* = 0.06). Internal rotation = 4 before G38—5 after vs. 4 before—7 after G42. Strength = 2 before—8 after G38 vs. 2 before—7 after G42.	-
Sebastià Forcada et al., 2014	Global score: 80 reverse vs. 56 partial. Pain: 14 reverse vs. 9 partial. Activity: 17 reverse vs. 12 partial*.	Flexion = 120° reverse 80° partial*. Abduction = 113° reverse vs. 79° partial. External rotation = 5 reverse vs. 3 partial. Internal rotation= 3 reverse vs. 3 partial (*p* = 0.9).	UCLA score: 29 reverse vs. 21 partial. DASH score: 17 reverse vs. 24 partial.
Boyer et al., 2017	Standard global score: 72 reverse vs. 72 partial.	Flexion = 109° reverse vs. 99.5° partial. Abduction = 99° reverse vs. 90° partial. External rotation = 21° reverse vs. 28° partial.	QuickDASH: 36 reverse vs. 78 partial.
Alentorn-Geli et al., 2014	Standard global score: 35 before vs. 57 after, no improvement in internal rotation. Greater difference before vs. after in reverse arthroplasties (standard global, front flexion and activity level)*.	-	Score SF-36 (quality of life): No difference.
Jorge-Mora et al., 2018	Global score: 53 cemented vs. 60 non-cemented/63 union vs. 45 non-union (difference of 15 points improvement).	Abduction = 92° cemented vs. 104° non-cemented/115° union vs. 68° non-union. Flexion = 92° cemented vs. 106° non-cemented/115° union vs. 69° non-union. Internal rotation = 35° cemented vs. 36° non-cemented/38° union vs. 31° non-union. External rotation = 17° cemented vs. 23° non-cemented/28° union vs. 5° non-union.	-
Sebastià Forcada et al., 2017	Last standard global assessment: 67 sequela vs. 78 fracture.	Strength = 2 sequela vs. 4 fracture. Flexion = 114° sequela vs. 127° fracture. Abduction = 104° sequela vs. 120° fracture. External rotation = 4 sequela vs. 5 fracture (*p* = 0.3). Internal rotation = 3 sequela vs. 3 fracture (*p* = 0.7).	Last assessment: Pain VAS 8 sequela vs. 8 fracture (*p* = 0.9), UCLA score: 26 sequela vs. fracture 29 fracture, QuickDASH score: 21.5 sequela vs. 25 fracture. Satisfied patients: 93%.
Lopiz et al., 2016	Global score: Lower in the older group.	No differences among groups at 24 months post intervention.	DASH score: 27 older group vs. 31 younger group (*p* = 0.1). Problems reported in EQ-5D: anxiety (38% older vs. 12% younger), pain/discomfort (23% vs. 12%), activity (38% vs. 0%), self-care (46% vs. 0%) and mobility (46% vs. 12%). Health status and quality of life EQ-VAS: 63 younger vs. 74 older.
Bonilla et al., 2012	Global score: 32 before—57.5 after Delta vs. 31 before—60 after Comprehensive.	Flexion (before—after) = 95°–130° Delta vs. 102°–132° Comprehensive. Abduction (before—after) = 86°–123° Delta vs. 98°–118° Comprehensive. Internal rotation (before—after) = 46%–44% Delta vs. 49%–61% Comprehensive. External rotation (before—after) = 43%–64% Delta vs. 51%–61% Comprehensive.	-
Izquierdo-Fernández et al., 2017	-	-	ASES score: 75 vs. 63. Length of stay: 5 vs. 6 days (*p* = 0.3).
Cáceres-Sánchez et al., 2015	Global score: 23 before—67 at 12 months.	External rotation = 26° before—67° after. Flexion = 74° before—135° after	There are no metrics for fractures before the intervention. EQ-VAS: 8 before vs. 2 at one-year follow-up. Satisfied patients: 100%.
Martinez et al., 2012	Global score: 28 before—58 after.	Flexion = 40° before vs. 100° after. Abduction = 41° before vs. 95° after. External rotation = 15° before vs. 35 after. Internal rotation = 25° before vs. 60° after.	Estimation of proximity to a normal back: 13% before to 56% after. Patient satisfaction: 86% satisfied or very satisfied.
Zafra et al., 2014	Global score: 23 before—65.5 after.	Flexion = 45° before vs. 117° after. Abduction = 39° before vs. 96° after*. Internal rotation (no difference). External rotation = 5° before vs. 15.5° after.	Cofield pain rating: 4.8 before vs. 1.77 after. Patient perception of improvement after the intervention: 95%.
García-Fernandez et al., 2015			Patients with periprosthetic fracture satisfied at the end of follow-up: 61%.
Alcobía-Díaz et al., 2017	Standardized Constant score: 36 before vs. 81 after.	Differences before vs. after: improved flexion (+15°/5°) and abduction (+10°/5°), not in rotation (-2°/0°). Goutallier classification: >2 grade.	Daily activity questionnaire: 20% limitation in shoulder function with low-demand tasks, 51% limitation with high-demand tasks. VAS pain = 3.5.
Torrens et al., 2016	Global score: 30 before—58 at one year. Pain: 5 before vs. 10 at one-year follow-up. Daily activities: 8 before vs. 14 at one-year follow-up.	Flexion = 4 before vs. 7 at one-year follow-up. Abduction = 4 before vs. 6 at one-year follow-up. External rotation = 3 before vs. 5 at one-year follow-up. Internal rotation = 4 before vs. 5 at one-year follow-up (*p* = 0.1). Strength = 2 before vs. 10 at one-year follow-up.	Patient perception of improvement: General = 80%, (minimum 8 points), Strength = 62 % (minimum 11 points), Anterior elevation = 73% (minimum 6 points), Lateral rotation = 73% (minimum 2 points), Internal rotation = 38% (minimum 2 points).
Torrens et al., 2017	Global score at 2-year follow-up: 54. Pain = 12. Daily activity = 15.	Flexion = 7, Abduction = 6, Lateral rotation = 5, Internal rotation =5, Strength = 5.	C-reactive protein; increase after surgery, peak on the 2nd day, recovery on the 14th.
Torrens et al., 2018	Global score at the end of follow-up: 61, 66 younger than 75 years vs. 57 older than 75 years. Pain: 12.	Metrics at the end of follow-up. Flexion = 7, Abduction = 6 Lateral rotation = 5. Internal rotation = 5, Strength = 7.	-
Delgado-Rodríguez et al., 2013		Flexion = 39° before vs. 84° after. External rotation = 13° before vs. 33° after. Internal rotation = 11° before vs. 31° after. Abduction = 32° before vs. 75° after.	QuickDASH score: 36% after. Pain VAS: 3.
Hernández-Elena et al., 2015	Global score: 63. Pain:14.	Abduction = 104°. Flexion = 106°. Internal rotation = 40°. External rotation = 46°.	
Villodre-Jiménez et al., 2016	Global score: 65. Best results in patients with arm lengthening intervention <20 mm.	Flexion = 124°. External rotation = 13°. Abduction = 95°.	UCLA scale: 27 points. QuickDASH: 32. Best results in QuickDASH in patients with arm lengthening intervention <20 mm. Patients with moderate to severe pain: 20%. Patients satisfied: 95%.
Andrés-Cano et al., 2014	Global score at the end of follow-up: 44 points.	Active abduction = 50°. Flexion = 70°. External rotation: 50°. Internal rotation: up to the lumbosacral joint.	QuickDASH: 24 points. Higher number of rehabilitation sessions = QuickDASH. Less operating time = QuickDASH. Pain EVA = 1 of 8.

G38: 38 mm glenosphere; G42: 42 mm glenosphere; -: no information available; UCLA: UCLA Shoulder rating Scale; DASH: Disabilities of the Arm Shoulder and Hand; SF-36: Short Form 36; ASES: American Shoulder and Elbow Surgeons; EQ-5D: Euro QoL 5D; VAS: Visual Analog Scale.

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
