# Peer review of "Safety and Effectiveness of Shoulder Arthroplasties in Spain: A Systematic Review"

_jcm, 2019, doi:10.3390/jcm8122063_

Round 1

Reviewer 1 Report

This is a systematic review of shoulder arthroplasties performed in Spain between 2003 and 2018. While the research question is of some interest to the orthopaedic community in Spain, it is slightly less relevant to an international audience. The methodology of the systematic review is sound. The data extraction of the included studies is thorough. The are however in my opinion a few weaknesses: 1) the authors give no explanation why the search was limited from 2003. Was shoulder arthroplasty not performed in Spain prior to that date?

2) I would recommend English language editing or at least review by a native speaker who is also a medical professional. 

3) I would recommend using a consistent nomenclature of reverse shoulder arthroplasty throughout the manuscript (various names are used,such as inverse, inverted, reverse).

4) the authors do not discuss all results. I would expect that all findings be discussed. Since this is a study about shoulder arthroplasty in Spain, one could for instance compare findings to other countries. 

5) lines 193-198: I do not understand this sentence.

6) the heterogeneous study population ought to be discussed more. For instance, I find it problematic to pool data (e.g. 15% complication rate) when the indication was proximal humeral fracture in some studies, while it was arthritis or cuff arthropathy in others.

Author Response

This is a systematic review of shoulder arthroplasties performed in Spain between 2003 and 2018. While the research question is of some interest to the orthopaedic community in Spain, it is slightly less relevant to an international audience. The methodology of the systematic review is sound. The data extraction of the included studies is thorough. There are however in my opinion a few weaknesses:

R: We want to thank the reviewer for the time spent reading our manuscript and the valuable comments given. Below you can find a list of the changes performed answering each specific comment made. Thank you again and we hope these revisions meet your expectations.

1)         the authors give no explanation why the search was limited from 2003. Was shoulder arthroplasty not performed in Spain prior to that date?

R: We want to thank the reviewer for this comment. We have now included an explanation of the exclusion criteria in the main document (please see lines 87 to 90 of the methods section). We want to highlight that we selected this timeframe because we feel the results of shoulder arthroplasties in these years is a reasonable representation of the continuous surgical improvement. Going further back and including literature and studies before 2003 could have led to an underestimation of the current global results of shoulder arthroplasty procedures given that surgical techniques are always evolving and improving.

2)         I would recommend English language editing or at least review by a native speaker who is also a medical professional.

R: We want to apologize for any mistakes made regarding the English language. Further language revision by a medical professional has been done.

3)         I would recommend using a consistent nomenclature of reverse shoulder arthroplasty throughout the manuscript (various names are used, such as inverse, inverted, reverse).

R: We apologize for this discrepancy. We have now used the same term (reverse shoulder arthroplasty) throughout the document in both text and tables.

4)         the authors do not discuss all results. I would expect that all findings be discussed. Since this is a study about shoulder arthroplasty in Spain, one could for instance compare findings to other countries.

R: Thank you for this comment. We have now included a more extensive comparison in the discussion section with the results from other countries (please see lines 205 to 218 and 232to 240 of the discussion). With this information added to the discussion, we feel our manuscript as a whole will be of increased interest to the international orthopedic community.

 5)         lines 193-198: I do not understand this sentence.

R: We apologize and after rereading the sentence it does seem wordy. We have rewritten it in an attempt to clarify and better explain the ideas discussed (please see lines 222 to 225 of the discussion).

6)         the heterogeneous study population ought to be discussed more. For instance, I find it problematic to pool data (e.g. 15% complication rate) when the indication was proximal humeral fracture in some studies, while it was arthritis or cuff arthropathy in others.

R: We thank the reviewer for this comment and we agree with it. We pooled the data to try to obtain a wider view of the current situation in Spain but have now included the problem of heterogeneity in the study populations of the literature included in this review as a limitation in the last paragraph of the discussion (please see lines 269 to 273 of discussion).

Reviewer 2 Report

This paper is a systematic review on the safety and effectiveness of shoulder arthroplasties, focused on the situation in Spain. The study was logically structured and the explanation of the selection of the publications was clear. 

Comments:

The authors explained clearly how the selection of the publications was made. In making this selection, some groups of applications were left out (e.g. studies including patients under 18 years of age, studies including patients with tumours, revisions etc - all explained in lines 87-91). However, in the discussion and in the conclusions (and in the abstract on line 34), this limitation is not taken into account anymore. E.g. lines 184-185 state that shoulder arthroplasty is an effective and safe procedure. It would be scientifically more sound to state that from the current review, this conclusion can be stated for a certain group of applications (enumerated in lines 138-140). Saying that shoulder arthroplasty is safe and effective in general, is not shown in the present review.  The introduction is well written and contains literature that can be expected there. On line 49, some complications are mentioned. It could be interesting to the reader to mention some more publications here, that go more into detail about possible complications. 

Smaller comments:

line 25: delete 'in' before on Table 2 and 3 mention for some implants the ODEP rating, and for others not (e.g. Lima SMR with a rating of 10A), is there a reason for that? line 163: delete the French sign at the end of the line

Author Response

This paper is a systematic review on the safety and effectiveness of shoulder arthroplasties, focused on the situation in Spain. The study was logically structured and the explanation of the selection of the publications was clear.

R: Thank you for the positive evaluation of our research. We would also like to acknowledge the time spent on revising our manuscript and the valuable comments given. We have made the changes proposed and have included a brief explanation for each specific change. We hope they meet your expectations and, again, thank you very much.

 Comments:

The authors explained clearly how the selection of the publications was made. In making this selection, some groups of applications were left out (e.g. studies including patients under 18 years of age, studies including patients with tumours, revisions etc - all explained in lines 87-91). However, in the discussion and in the conclusions (and in the abstract on line 34), this limitation is not taken into account anymore. E.g. lines 184-185 state that shoulder arthroplasty is an effective and safe procedure.

R: Thank you for this valuable suggestion. In an attempt to be more cautious and more precise when we discuss the generalizability of the results obtained, we have now included the limitation related to the populations selected in the discussion and conclusions sections and we have also mentioned this limitation in the abstract (please see lines 204 to 209 of the discussion, lines 275 and 279 of the conclusions and lines 36 and 37 of the abstract).  

It would be scientifically more sound to state that from the current review, this conclusion can be stated for a certain group of applications (enumerated in lines 138-140). Saying that shoulder arthroplasty is safe and effective in general, is not shown in the present review. 

R: Thank you very much for this comment. We have now focused the conclusions on the group of patients for whom the study results could be applicable and we have tried to be more specific when discussing the safety and effectiveness of shoulder arthroplasties in Spain (please see lines 204 to 209 of the discussion and 275 and 279 of the conclusions sections).

The introduction is well written and contains literature that can be expected there. On line 49, some complications are mentioned. It could be interesting to the reader to mention some more publications here, that go more into detail about possible complications.

R: Thank you for this suggestion, we have now included more references (please see references 12 to 14). Additionally, we have listed some of the most frequent complications found in other contexts (please see lines 55 to 59 of the introduction section).

Smaller comments:

line 25: delete 'in' before on Table 2 and 3 mention for some implants the ODEP rating, and for others not (e.g. Lima SMR with a rating of 10A), is there a reason for that? line 163: delete the French sign at the end of the line

R: We apologize for these mistakes in line 25 and for the mistake in line 163 and have corrected them. Regarding the ODEP implant ratings, we have reviewed the classification and included all scores and corrected the Delta Xtend score. Again, we apologize for these errors.

Round 2

Reviewer 1 Report

The authors have done a very nice job improving the manuscript. I recommend publication.